# Effects of Sulforaphane-Induced Cell Death upon Repeated Passage of Either P-Glycoprotein-Negative or P-Glycoprotein-Positive L1210 Cell Variants

**DOI:** 10.3390/ijms231810818

**Published:** 2022-09-16

**Authors:** Anna Bertova, Szilvia Kontar, Zoltan Polozsanyi, Martin Simkovic, Zuzana Rosenbergova, Martin Rebros, Zdena Sulova, Albert Breier, Denisa Imrichova

**Affiliations:** 1Institute of Molecular Physiology and Genetics, Centre of Biosciences, Slovak Academy of Sciences, Dúbravská cesta 9, 840 05 Bratislava, Slovakia; 2Institute of Biochemistry and Microbiology, Faculty of Chemical and Food Technology, Slovak University of Technology in Bratislava, Radlinského 9, 812 37 Bratislava, Slovakia; 3Institute of Biotechnology, Faculty of Chemical and Food Technology, Slovak University of Technology in Bratislava, Radlinského 9, 812 37 Bratislava, Slovakia

**Keywords:** sulforaphane, multidrug resistance, ABCB1 transporter, autophagy, LC3B protein, cyclins, cyclin dependent kinases, apoptotic proteins, monodansylcadaverine

## Abstract

The expression of the membrane ABCB1 transporter in neoplastic cells is one of the most common causes of reduced sensitivity to chemotherapy. In our previous study, we investigated the effect of a single culture of ABCB1-negative (S) and ABCB1-positive variants of L1210 cells (R and T) in the presence of sulforaphane (SFN). We demonstrated that SFN induces the onset of autophagy more markedly in S cells than in R or T cells. In the current study, we focused on the effect of the repeated culture of S, R and T cells in SFN-containing media. The repeated cultures increased the onset of autophagy compared to the simple culture, mainly in S cells and to a lesser extent in R and T cells, as indicated by changes in the cellular content of 16 and 18 kDa fragments of LC3B protein or changes in the specific staining of cells with monodansylcadaverine. We conclude that SFN affects ABCB1-negative S cells more than ABCB1-positive R and T cells during repeated culturing. Changes in cell sensitivity to SFN appear to be related to the expression of genes for cell-cycle checkpoints, such as cyclins and cyclin-dependent kinases.

## 1. Introduction

Despite great progress in the rational development of anticancer drugs, which has resulted in the targeted treatment of individual subtypes of oncological diseases, these diseases are still a major medical problem that leads to numerous deaths. For example, nearly 10 million people died of cancer worldwide in 2020, accounting for almost one in six deaths [1]. This is due to multiple reasons, among which the development of neoplastic tissue and cell resistance to substances used to suppress or eliminate disease progression play an important role [2]. Cells are capable of developing resistance to multiple drugs (MDR) with different structures and mechanisms of action, which can be caused by (i) changes in the number and sensitivity of drug targets; (ii) drug modification with phase 1 and phase 2 detoxification enzymes; (iii) increased drug efflux by membrane transporters, which represents detoxification phase 3; (iv) streamlining damage-repair mechanisms, including DNA repair; (v) changes in the regulation of cell-death programs; and (vi) changes in cell cycle and proliferation regulation (reviewed in Reference [3]). The most commonly observed molecular cause of MDR is massive cellular drug efflux mediated by drug transporters that are predominantly members of the ABC protein family, in particular ABCB1 transporter (also known as P-glycoprotein), ABCC1 (also known as multidrug resistance-associated protein) and ABCG2 (also known as breast cancer resistance protein) [4], which can adequately protect cells from the toxic effects of a wide range of drugs. Of these, the ABCB1 transporter is the most common [5]. The coupled changes in protein expression that accompany ABCB1 overexpression may limit cell entry into cell death even when substances other than its substrates are used [6]. Therefore, it is important to look for substances that are also effective on cells that have developed MDR mediated by ABCB1.

Plant isothiocyanates (ITCs) appear to be promising agents for preventing and suppressing cancer progression [7,8]. The cytotoxic and carcinostatic activity of some ITCs (e.g., sulforaphane (SFN), which is present in cruciferous vegetables; and allyl isothiocyanate (AITC), which is present in horseradish and wasabi) was described as early as 1968, using HeLa cells [9]. However, there is currently a renewed interest in plant ITCs due to new data on their chemopreventive effects against various chronic diseases, such as neoplastic, cardiovascular, neurodegenerative and metabolic diseases [10,11,12,13]. ITCs can modulate the induction, transcription and function of a wide range of metabolic and regulatory proteins by chemical modification of the nucleophilic functional groups of bio(macro)molecules [14,15]. However, the antioxidant effects of ITCs could also take part in their biological effects [16]. The mechanism of the antioxidant effect of SFN may be related to the activation of nuclear factor erythroid-2-related factor 2 (also known as Nrf2, a transcription factor of the cellular response to oxidative stress), and this may be related to the promotion of autophagy [17]. Similar to other cytotoxic agents, ITCs preferentially attack proliferative and transcriptionally highly active cells, including neoplastically transformed variants [18].

Cruciferous vegetables contain glucosinolates as sulfur-containing storage compounds [19] from which ITCs, as well as other active substances (epithionitrile, thiocyanates, nitriles, oxazolidine-thione and other less-common compounds [20]), are released by the myrosinase (Myr, a β-thioglucoside glucohydrolase, EC 3.2.1.147) reaction [21]. Although glucoraphanin (GR, a glucosinolate precursor of SFN) is thought to have no direct effect on neoplastic cells [22], we decided to compare the effect of SFN with GR in the absence and presence of active recombinant Myr.

The characterization of the effect of two plant ITCs (AITC and SFN) on cell proliferation, cell cycle, onset, progression and type of cell death, as well as the expression of the protein players involved in its regulation, has been described in a previous paper [23]. We were able to identify LC3B protein-dependent autophagy, leading to cell death, as the predominant cell-death mechanism, which was more pronounced in ABCB1-negative cells. However, in our previous study [23], the effect of ITCs was observed after one passage of cells in a medium containing ITCs. In this work, we focused on investigating how cells respond when exposed to repeated passages in an SFN-containing medium.

## 2. Results

In our experiments, we used the following three variants of murine leukemia cells: (i) the original ABCB1-negative L1210 cells (S), (ii) ABCB1-positive cells (R) in which ABCB1 expression was induced by vincristine [24] and (iii) ABCB1-positive (T) cells in which ABCB1 expression was induced by transfection with the human gene encoding ABCB1 [25]. We primarily aimed to compare the cytotoxic effects of SFN, GR and GR transformed by Myr on the S variant of L1210 cells. In another set of experiments, we examined the effect of repeated cultures of S, R and T cells in the presence of SFN.

### 2.1. Comparison of the Effects of SFN and GR on Cell Death

We performed these experiments by using parental S cells, which we subjected to a single passage in the presence of GR (10 and 50 μM), SFN (10 and 50 μM), ascorbic acid (AA, a noncompetitive Myr activator [26], 10 μM) and recombinant Myr (prepared and characterized in Reference [27], 0.05 U/mL), each alone, and GR in combination with Myr and AA at the same concentration (Figure 1). 

While SFN at concentrations of 10 and 50 μM induced a significant cell-death effect, reaching a 50% and a more than 90% decrease in surviving cells, respectively, GR appeared to be ineffective (Figure 1A). We hypothesized that transformation of GR in a Myr reaction would provide sufficient SFN for induction of cell death. Although the Myr reaction produces an unstable aglycone from GR, which spontaneously hydrolyzes to different products depending on the conditions, at pH ≈ 7.0, SFN should be the dominant product, and in acidic environments, sulforaphane nitrile is formed (Figure 1B) [28,29].

Myr alone (0.05 U/mL) did not affect cell viability (Figure 1A). However, in the case of AA, which at concentrations less than 10 μM did not induce significant cell death, massive cell-death effects above this threshold were observed (Appendix A). Therefore, it was necessary to use AA at a concentration of 10 μM in further experiments. When GR 10 μM and 50 μM were applied together with Myr (0.05 U/mL) and AA (10 μM), we observed a 37% and 67% decrease in cell viability, respectively (Figure 1A). These results clearly demonstrate that SFN alone, and not its glucosinolate precursor GR, causes a significant cell-death effect.

### 2.2. Cell Proliferation and Specific Gene Expression in S, R and T Cells Pretreated with SFN

After one passage in the presence of SFN (at concentrations of 0–10 μM), the cells were harvested by centrifugation and seeded for the next passage in the absence of SFN. For these cells, we determined the total amount of cell growth after 48 h (Appendix A) and the time dependence of cell growth after 4, 8, 12, 24 and 48 h (Appendix A).

The number of viable cells was counted in the CASY Model TT-cell counter (see Section 4.2). We observed a decrease in the number of cells only in S cells and only if the cells were precultured at the highest concentration of SFN (i.e., 10 μM; see Appendix A). As in our previous study [23], the kinetics of cell growth over 48 h showed a linear dependence in the semilogarithmic plot. When monitoring the growth kinetics, the growth rate slowed only in S cells that were precultured at the highest concentration of SFN. The characteristic growth kinetics of S, R and T cells precultured in the absence of SFN and in the presence of 10 μM SFN are documented in Appendix A. In these plots, the slope of the straight line is the rate constant, k, which is slightly reduced only for S cells precultured at 10 μM SFN to a value of 0.0147 s^−1^ compared to the control value in the absence of SFN 0.0184 s^−1^. For all other cases, the rate constant varied in the interval 0.0174 < k < 0.0189.

We further analyzed the expression of the *Ccnd1* (cyclin D1), *Cdk6* (cyclin-dependent kinase 6), *Ccne1* (cyclin E1), *Cdk2* (cyclin-dependent kinase 2), *Bcl2* and *Bax* genes by RT-PCR in cells cultured for 48 h in the presence of SFN (0–10 μM, Figure 2) and in cells in which this preculturing was followed by culture in the absence of SFN (Figure 3).

Cyclin D1 gene expression was increased in R cells cultured in the absence of SFN compared to S cells (Figure 2 and Figure 3). The increase in the expression of this cyclin in T cells was less pronounced; however, statistically significant differences were observed. These results are consistent with the findings described in a previous study [30]. In S cells, culturing in the presence of SFN does not induce changes in cyclin D1 expression, in contrast to R cells, where a concentration-dependent increase in expression is visible (Figure 2). Higher levels of cyclin D1 were maintained in R cells even after further culture in SFN-free medium, but the increasing trend with increasing SFN concentration from the previous culture disappeared (Figure 3).

In T cells, we did not observe a clear dependence of the increase in cyclin D1 expression on SFN concentration. The gene for cyclin-dependent kinase 6 (CDK6), the enzyme that is activated by cyclin D1 [31], was overexpressed in R and T cells compared to S cells (Figure 2 and Figure 3), similar to what was described previously [30]. This increased expression seemed to persist even after culturing with SFN (Figure 2) or after additional culturing without SFN (Figure 3), but we did not observe concentration-dependent changes in the expression of this kinase. As in a previous study [30], cyclin E1 is overexpressed in R and T cells compared to S cells (Figure 2 and Figure 3). SFN caused a further increase in the expression of this cyclin (Figure 2), but this increase was lost after subsequent cultivation in the absence of SFN (Figure 3). However, we did not observe significant changes in the expression of the cyclin-dependent kinase 2 (CDK2) gene, an enzyme that is activated by cyclin E1 [32], either compared to R and T cells or after SFN administration. The expression of the gene for the antiapoptotic proto-oncogene Bcl-2, which is often responsible for the resistance of cells to chemotherapy [33], was significantly increased in ABCB1-positive R and T cells compared to S cells (Figure 2). In contrast, the expression of the gene encoding the Bax protein (proapoptotic protein of the Bcl-2 family [34]) was approximately the same in R and T cells as in S cells (Figure 2). We have described the expression levels of both Bcl-2 family proteins in S, R and T cells in previous papers [23,35]. Culturing the cells in the presence of SFN did not cause significant changes in the expression of either protein, except for a slight but significant upregulation of Bcl-2 after culturing the cells in medium containing 5 μM SFN. This effect was lost after subsequent culturing of the cells in SFN-free medium (Figure 3). SFN did not induce any significant changes in Bax protein expression during 48 h of incubation (Figure 2), but after subsequent incubation in the absence of SFN, Bax expression decreased slightly in R cells originally cultured in 7.5 μM SFN medium.

### 2.3. Effects of Repeated Cultivation of S, R and T Cells in SFN-Containing Medium on Cell Growth and Gene Expression

In the first part of these experiments, the cells were passaged six times in the presence of SFN (0–15 μM) for 48 h. The number of cells after each passage was determined by using a CASY Model TT-cell counter (see Section 4.2). To calculate the median inhibitory concentrations (IC_50_), the dose-response curves (Appendix A) were fitted according to Equation (1) (see Section 4.4). The IC_50_ values obtained are documented in Figure 4. Growth suppression with SFN occurred in S cells during the first and second passages with a similar IC_50_ value (in the range of 7–8 μM, Figure 4). For R and T cells, we obtained statistically significantly higher values of this parameter after the first passage (11 μM and 13 μM for R and T cells, respectively).

These values increased even more after the second passage. From the third passage, IC_50_ values decreased in all three cell variants (Figure 4) and reached a value in the range of 3.5–3.8 μM for S cells, 6.5–7.2 μM for R cells and 7.4–7.9 μM for T cells after the fourth and sixth passages. Thus, repeated culturing of cells in the presence of SFN caused an increase in cell sensitivity corresponding to a decrease in IC_50_ of approximately 50% for S cells, 65% for R cells and 55% for T cells.

Next, we determined the expression of the *Ccnd1*, *Cdk6*, *Ccne1*, *Cdk2*, *Bcl2* and *Bax* genes in cells after six repeated passages in SFN-containing medium (0–10 μM, Figure 5). The number of S cells after six passages in the presence of SFN at a concentration of 10 μM was significantly reduced, and after isolation of total RNA, we did not obtain enough quality polyribonucleic acid. This caused an attenuated amount of PCR product for GAPDH and amounts of PCR products for genes of interest at the limit of quantification as measured by RT-PCR (Figure 5A). Therefore, in the groups of S cells passaged 6 × in the presence of 10 μM SFN, we always obtained a reduced level of the PCR product of the respective genes. In the case of ABCB1-positive R and T cells, sufficient total RNA was obtained for reliable determination after six passages, even at a concentration of 10 μM SFN. After six consecutive passages in the absence of SFN, we did not observe significant differences in the content of the two cyclins (D1 and E1) when comparing S and R cells (Figure 5). However, after six passages in the presence of higher concentrations of SFN (5–10 μM), the expression of both cyclins increased in R cells when compared to S cells. In T cells, we observed lower expression of cyclin D1 but increased the expression of cyclin E1 after six consecutive passages in the absence of SFN. In T cells, changes in the expression of both cyclins after six passages in SFN-containing media were less pronounced than in R cells. In R and T cells, we observed a decrease in CDK6 after six consecutive passages in the absence of SFN compared to S, a difference that persisted even when passaged in the presence of SFN, with the exception of T cells passaged six times in medium containing 10 μM SFN. CDK2 expression appeared to be similar in S, R and T cells and did not change with SFN concentration after six passages. The gene for the proto-oncogene Bcl-2 was upregulated after six passages in the absence of SFN in R and T cells compared to S cells, but this difference was less pronounced than that observed after a single passage (Figure 2). Six repeated passages in the presence of SFN slightly elevated the expression of this gene. The expression of the gene for the proapoptotic Bax protein was almost identical in S, R and T cells and was independent of SFN concentration.

### 2.4. Changes in the Cellular Content of Cyclin B, Cyclin E and LC3B Protein Fragments Induced by Repeated Passaging in SFN-Containing Medium

SFN has been described to induce cell-cycle arrest in the G2/M phase of the cell cycle and the gradual entry of cells into apoptosis [36]. In our previous paper, we observed an increase in the cell fraction in the G0/G1 phase of the cell cycle in R and T cells, but in S cells the cell cycle was not altered by SFN [23]. Based on a double-staining assay with Annexin-V/propidium iodide, we detected only insignificant amounts of cells in apoptosis and thus labeled only with Annexin V. However, in SFN-treated cells, we detected an increased presence of 16 and 18 kDa fragments of LC3B protein and an increase in labeling of cells with monodansylcadaverine (MDC) [23], which is localized in autophagic vesicles and is known as an indicator of autophagy [37]. Therefore, we focused our experiment on determining the cellular content of cyclin E (active in the G1 to S phase transition of the cell cycle [38]) and cyclin B active in the M phase of the cell cycle [39]. For these experiments, we changed the procedure of repeated passages in the presence of SFN, since after six passages in the presence of 10 μM SFN, we obtained few cells and little material. Therefore, we reduced the number of passages to four but increased the highest concentration of SFN to 15. In experiments measuring dose-response curves (Appendix A), we verified that, even if we had only approximately 10% of the cells after four passages, we could isolate enough material from them for further analyses.

In our previous work, we showed that there is no difference in the cellular mRNA content for the *Ccnb1* gene encoding cyclin B1 between S, R and T cells (which we documented in the Appendix A attached in our previous paper [35]). As we initially had no information about the effect of SFN on the expression of the *Ccnb1* gene, we determined its expression in S, R and T cells after one and four passages in SFN-containing medium (0–15 μM, Appendix A). However, we did not observe significant changes in the quantity of PCR products for this gene between S, R and T cells after either one or four passages with SFN.

S and T cells after one passage in the absence of SFN had approximately the same cyclin B cell content, which was slightly lower than that in R cells (Figure 6A,B). One passage of S cells in the presence of SFN (5–15 μM) did not affect the cellular content of this cyclin. In R cells, however, we observed a strong decrease in cell content, but only after passage with the highest concentration of SFN (15 μM SFN; Figure 6A,B). T cell passage in 5 μM SFN resulted in a doubling of the content of cyclin B in cells, and a slight increase was observed even after passage in 10 μM SFN. Only passage in 15 μM SFN induced a decrease in the cellular content of this cyclin.

The protein cell content for cyclin E was slightly higher in S cells than in R and T cells, and after one passage in SFN medium, its content decreased in a concentration-dependent manner (Figure 6A,B). The change in the cellular content of the 16 and 18 kDa LC3B protein fragments was particularly pronounced. Both fragments were less abundant in R and T cells, with the 18 kDa fragment being more prominent (Figure 6A,B). One passage of cells in SFN-containing medium provided a concentration-dependent increase in both fragments in S cells and, conversely, a decrease in R and T cells.

After four passages of S cells in SFN medium (15 μM), cyclin B and E cell levels were significantly reduced (Figure 6C,D). When the SFN concentration was 5 μM during the passages, the amount of cyclin B increased, and the amount of cyclin E remained stable at a level similar to the control condition without SFN. In R and T cells, the level of cyclin B increased after four passages in the presence of 5 and 10 μM SFN compared to the control in the absence of SFN (Figure 6C,D). In these cell variants, the level of cyclin B fell below the control value only after passages in medium containing 15 μM SFN. The level of cyclin E in T cells decreased only slightly after four passages in the presence of SFN, even at the highest concentration. In R cells, such a slight decrease occurred when passaged in 5 and 10 μM SFN medium, but when passaged at 15 μM, cyclin B levels were reduced to one quarter of the control value. The most significant differences between ABCB1-negative S cells and ABCB1-positive R cells were observed for the cell levels of LC3B protein fragments. In S cells, the levels of both fragments increased with SFN concentration, and in R and T cells, they stabilized closer to the control values and were always significantly lower than in S cells (Figure 6C,D). Because LC3B fragments are involved in autophagy, including autophagosome expansion and fusion, their status can serve as a biomarker for autophagy [40]. Therefore, SFN, especially during repeated passages, induced autophagy preferentially in S cells. In R and T cells, we observed the presence of LC3B proteins, but without an apparent dependence on the concentration of SFN, either at one or four passages.

We also decided to analyze more massive autophagy in S cells by labeling living cells in situ with MDC and visualizing the labeled structures with a confocal microscope (Figure 7). MDC is a lysosomotropic compound useful for the identification of autophagic vesicles by fluorescence microscopy and, in addition, for the evaluation of the induction of autophagy by the accumulation of MDC-labeled vacuoles [41]. MDC is a substrate of the ABCB1 transporter and is therefore efficiently effluxed in ABCB1-positive cells, thus eliminating autophagic vesicle labeling (we documented this in the Appendix A attached in a previous paper [23]). In this work, we addressed this barrier by using the high-affinity noncompetitive inhibitor of the ABCB1 transporter tariquidar (TQR), which, at a concentration of 500 nM, ensured sufficient retention of MDC in R and T cells and reliable visualization of autophagic vesicles. This concentration of TQR does not affect the viability of either ABCB1-negative S or ABCB1-positive R and T cells [42]. Therefore, in the following experiments, we used TQR at this concentration (Figure 7).

After four passages in ABCB1-negative S cells, we observed the growth of green-labeled autophagic vesicles by using 5 μM SFN. In contrast, for ABCB1-positive cells, such an increase was present only if the highest concentration of SFN (15 μM) was used.

## 3. Discussion

The development of multidrug resistance is a serious obstacle to successful treatment and is increasing the search for agents that are also able to attack ABCB1-overexpressing cells. Such substances also appear to include plant ITCs such as SFN or AITC [23].

These substances are stored in plants in the form of glucosinolates, from which they can be released by the enzyme Myr (β-thioglucoside glucohydrolase, EC 3.2.1.147) [45]. Glucosinolates are considered inactive but unstable aglycones originating from the Myr reaction and are converted to ITCs, nitriles indoles and others, which have significant biological effects [11]. Recently, sinigrin and glucotropaeolin have been shown to produce only negligible antiproliferative effects on HT29 cells [46]. However, when Myr was added to each glucosinolate, the effects observed approached the effects of allyl and benzyl isothiocyanates released from them by Myr. We also chose this approach to compare the effects of GR and SFN (Figure 1) on the proliferation of S cells. As already described, Myr requires activation by AA [26].

The recombinant Myr used in our experiments achieves optimal activation at an AA concentration of 1 mM [47]. However, when AA was added to the medium at a concentration exceeding 10 μM, it significantly suppressed the growth of S cells (Appendix A). The recombinant Myr used at this AA concentration can reach 10–20% of the activity observed at the optimal AA concentration of 1 mM [47]. The optimal concentration of AA could depend on the type of Myr and how the enzyme was obtained, naturally or recombinantly, as the matured enzyme may differ in posttranslational treatment, even if microbial eukaryotes have been used as a producer. For example, Myr isolated from germinated *Lepidium sativum* seeds achieved optimal activation with AA at concentrations ranging from 20 to 30 μM [48]. 

In this experiment, we used SFN and GR at concentrations of 10 and 50 μM, which were chosen for the following reasons. In a previous paper, we showed that when measuring cell death by reduction of tetrazolium salt to formazan at low sulforaphane concentrations, an increase in formazan production is paradoxically observed [23]. However, the corresponding increase in cell number detected by counting on cell counter was missing. Such a phenomenon occurs due to metabolic hyperactivation, which can also be observed after UV irradiation [49]. Similar increases in formazan signal after SFN application have also been observed by others [50]. In the current study, SFN at a concentration of 1 μM caused a significant increase in formazan signal compared to the control (Appendix A). Conversely, the SFN at concentrations 10 and 50 μΜ caused a significant monotonic concentration-dependent decrease in formazan signal. If the number of viable cells after passaging with SFN is detected on cell counters, its monotonic decrease with SFN concentration in the concentration range of 1–50 μM was obtained (Appendix A). Thus, at SFN concentrations in the range of 10–50 μM, the decrease in formazan signal in S cells is monotonic and is accompanied by a proportionally decreasing number of viable cells (Appendix A). Therefore, we chose SFN concentrations of 10 μΜ and 50 μΜ as close to the IC50 and highly toxic concentrations, respectively, and also used GR at these concentrations. Furthermore, we are outside the concentration range in which SFN-induced metabolic hyperactivation of cells would cause inaccuracies in the measurement of cell viability by the MTT assay. 

In contrast to GR, which did not cause significant changes in S-cell proliferation even at a concentration of 50 μM, SFN induced significant cell-death effects even at a concentration of 10 μM (Figure 1). In contrast, when 10 μΜ GR alone was used, a slight increase of formazane signal was observed but was not statistically significant. However, when recombinant Myr was added to the medium together with GR (10 μM and 50 μM), a significant cell-death effect of 37% and 67% growth inhibition was observed. This indicates that SFN, and not GR, causes cell death; this result is consistent with the findings of Kolodziejski et al. [46] reported for other glucosinolates/ITCs. This may be due to the absence of the reactive isothiocyanate group, which is protected in GR by bound thioglucose. However, we should consider how glucosinolates have a more hydrophilic nature than ITCs do, and this may reduce their bioavailability [46].

To be effective, substances must act at a sufficient concentration for a certain time. To observe a longer-term effect of SFN on cells than the period of a single passage, we subjected them to repeated culturing in the presence of this ITC.

SFN in the first passage inhibited the growth of S, R and T cells with an IC_50_ in the range of 5–15 µM, with ABCB1-negative S cells appearing to be more sensitive than ABCB1-positive R and T cells (Appendix A and Figure 4).

We also investigated whether the expression of the ABCB1 transporter changes during SFN application. There is ambiguous information in the literature on this issue. On the one hand, inhibition of the pregnane X receptor [51] or β-STAT3 (phosphorylated signal transducer and activator of transcription 3) [52] by SFN may suppress or reduce ABCB1 regulation, respectively. On the other hand, activation of the transcription factor NRF2 with SFN may lead to upregulation of ABCB1 [53]. However, it has also been documented that SFN does not induce changes in ABCB1 expression in different tumor lines [54]. Therefore, we examined the expression of the *Abcb1* gene in our cells. Our results showed that there was no visible change in *Abcb1* gene expression even after a single passage in SFN-containing medium or after combination of this passage with subsequent passage without SFN, as well as after six consecutive passages in the presence of SFN (Appendix A). Therefore, we conclude that passages of L1210 cell variants do not lead to induction of ABCB1 expression in S cells or modulation of its expression in R and T cells.

To better visualize the changes in gene expression (Ccnd1, Cdk6, Ccne1, Cdk2, Bcl2 and Bax; Figure 2, Figure 3 and Figure 5), we summarize the data in Table 1.

R- and T-cell variants had increased gene expression for *Ccne1* and *Bcl 2* compared to S cells (Table 1). This persisted after one passage with SFN (Figure 2), after one passage in the presence of SFN and after a subsequent passage without SFN (Figure 3), and after six consecutive passages in the presence of SFN (Figure 5). Cyclin E1 knockdown restores the sensitivity of sorafenib-resistant hepatocellular carcinoma cells to this drug [55]. In addition, cyclin E1 induction has been identified as a cause of temozolomide resistance in glioblastoma cells. Thus, increased cyclin E1 expression may be associated with drug resistance of cells [56]. Surprisingly, the cellular content of cyclin E protein (Figure 6) appears to be lower in R and T cells than in S cells, in contrast to the higher amount of mRNA encoding this protein in R and T cells than in S cells shown here in Figure 2 and described in previous papers, as well [30,35]. This is an example of a discrepancy between the amounts of protein product and coding mRNA that can be observed [57]. This discrepancy is the result of the fact that proteins and RNA represent different steps in the multistep transition of genetic information to functional proteins, in which these are dynamically generated and degraded [58]. After one passage of cells in the presence of SFN, we detected an increase in *Ccne1* expression in S and R cells. However, overexpression of this gene in T cells was observed only at the highest concentration of SFN (Figure 2, Figure 3 and Figure 5, respectively; summarized in Table 1). Subsequent cell passage in the absence of SFN appeared to attenuate the increase in the expression of this cyclin, especially in R cells. Even after six passages of R and T cells in SFN-containing medium, they retained increased *Ccne1* gene expression compared to S cells (Figure 5). The expression of the cyclin-dependent kinase 2 (*Cdk2*) gene, which is activated by cyclin E [32], was approximately the same in S, R and T cells and was not affected by cell passage in SFN-containing medium (Figure 2, 3 and 5 and summarized in Table 1). Increased cyclin D gene expression was observed after one or two passages in the absence of SFN in R cells, which thus differed significantly from S and T cells. After six consecutive passages, the difference between S and R cells disappeared, and we even observed lower expression of the *Ccnd1* gene in T cells than in S cells (Figure 5). After one passage of the cells in the presence of SFN, the expression of the *Ccnd1* gene increased in R and T cells and remained unchanged in S cells. When this passage was followed by a passage in the absence of SFN, the differences in *Ccnd1* expression in R and T cells were attenuated. Increased cyclin D gene expression was observed after one or two passages in the absence of SFN in R cells, which thus differed significantly from S and T cells (Table 1). After six consecutive passages, the difference between S and R cells disappeared, and we even observed lower expression of the *Ccnd1* gene in T cells than in S cells. After one passage of the cells in the presence of SFN, the expression of the *Ccnd1* gene increased in R and T cells and remained unchanged in S cells (Figure 2, Table 1). When this passage was followed by a passage in the absence of SFN, the differences in *Ccnd1* expression in R and T cells were attenuated (Figure 3). Even after six consecutive passages of R and T cells in SFN-containing medium, they retained increased *Ccne1* gene expression compared to S cells (Figure 5). The expression of the cyclin-dependent kinase 6 (*Cdk6*) gene, which is activated by cyclin D [31], was approximately the same in S, R and T and was not affected by cell passage in SFN (Figure 2, Figure 3 and Figure 5 and summarized in Table 1). Thus, an increase in the expression of genes representing the checkpoints of cell transition from G1 to S phase of the cell cycle in ABCB1-positive R and T cells was observed. The importance of the CDK4/6 checkpoint is underlined by the great attention currently being paid to the search for small molecules capable of inhibiting it [59]. However, cells are able to develop resistance to these inhibitors by various mechanisms, including increased expression or activity of CDK2 or suppression of levels of endogenous CIP/KIP protein inhibitors of CDKs (p21 and p27) [60,61]. Because inhibition of CDK2, 4 and 6 exhibits potent antitumor activity [62], increased regulation of proteins involved in the formation of their mediated checkpoints is expected to increase tumor cell survival. This may be at least partly responsible for the slightly higher resistance of R and T cells to SFN.

ABCB1-positive R and T cells had significantly increased levels of *Bcl2* gene expression compared to their ABCB1-negative counterparts (Figure 2, Figure 3 and Figure 5 and Table 1). This difference was maintained even after single or multiple cell passages in SFN-containing medium. We previously described the upregulation of Bcl-2 in L1210 cells expressing ABCB1 compared to ABCB1-negative counterpart cells at both the mRNA and protein levels [23,35,63]. The combined overexpression of both ABCB1 and Bcl-2 is a phenomenon that may increase the ability of ABCB1-positive MDR cells to survive chemical stress induced by anticancer drugs [6]. When their expression in the leukemic blasts of AML patients was monitored, it was shown that neither overexpression of ABCB1 alone nor Bcl-2 alone altered the overall survival of patients during treatment [63]. However, in patients with AML blasts overexpressing both of these proteins together, a significant reduction in their overall survival was observed. On the other hand, no changes in proapoptotic *Bax* gene expression were observed when we compared S, R and T cells (Figure 2, Figure 3 and Figure 5 and Table 1). We confirmed this result several times at the mRNA and protein levels [23,35,63]. The prevalence of antiapoptotic gene expression (e.g., *Bcl2*) over proapoptotic genes (e.g., *Bax*) suggests enhanced antiapoptotic stimuli [64]. Thus, a reduction of R and T cells entering apoptosis compared to S cells can be expected. No significant changes in the *Bax* gene were observed even when the cells were passaged in SFN-containing medium (Figure 2, Figure 3 and Figure 5 and Table 1). Based on these data, it can be assumed that the predominance of Bcl-2 expression over Bax is a factor that plays a role in the slightly lower sensitivity of R and T cells compared to S cells (Appendix A and Figure 4) to SFN in single or repeated passages.

The cyclin B1 cell levels were higher after one passage in R cells than in S and T cells, but after four repeated passages, we observed an increase in its level in T cells (Figure 6, Table 2). The cyclin E1 levels were slightly lower in R and T cells than in S cells. However, after four repeated passages in medium containing 10 μM SFN in R and T cells, the levels of both cyclins were slightly decreased (Table 2) but still relatively high compared to those in S cells, where they largely decreased (Figure 6). After four passages in medium containing 15 μM SFN, the protein levels of these cyclins also largely decreased in R and T cells.

We found the most significant differences in the cell content of both 16 and 18 kDa LC3B protein variants, which were lower in R and T cells than in S cells (Table 2). In addition, the levels of these protein variants remained relatively low even after passages in media containing SFN (5–15 µM) in R and T cells, in contrast to S cells, where they increased in a concentration-dependent manner (Figure 6). As LC3B proteins are localized in autophagic structures, their detection by LC3 by immunoblotting or immunofluorescence is a reliable method for monitoring autophagy and processes associated with autophagy, including autophagic cell death [65]. In this regard, S cells treated with SFN by either a single or repeated passage are more prone to entering autophagy than R and T cells. We investigated this by monitoring the induction of autophagy associated with the formation of autophagic vacuoles, using the lysosomotropic compound MDC as a fluorescent marker for microscopy [41]. In S cells passaged four times in medium containing SFN (5-15 μM), we observed an increase in MDC labeling in a concentration-dependent manner (Figure 7). For R and T cells, we observed an increase in MDC labeling above the control (absence of SFN) level only at 15 μM SFN, while the labeling at the two lower SFN concentrations was similar to the control. The consequences of autophagy in its induction by SFN must be properly assessed. Autophagy is a catabolic process that accelerates in cells in response to stress conditions (e.g., oxidative stress caused by toxic substances or irradiation) and allows cells to survive by maintaining cellular homeostasis by degrading and recycling intracellular components [66]. If the cell is successful in this process, it can continue to proliferate if it does not die. It must be assumed that apoptosis and autophagy are possible alternatives to the cellular response after administration of a toxic agent with which the former clearly leads to cell death and the latter either allows repair of damage and survival or leads to death. Such cell behavior after SFN-induced stress is supported by the research of other authors. In human prostate cancer cells PC-3 and LNCaP, SFN-induced autophagy has been associated with autophagosome upregulation linked with an increase in the cell content of 18 and 16 kDa proteolytic fragments of the LC3B protein [67]. Treatment of cells with a specific autophagy inhibitor (3-methyladenine) attenuated the localization of LC3 to autophagosomes but enhanced the cytosolic release of cytochrome c with subsequent apoptotic cell death. SFN inhibits cell proliferation and induces apoptosis through caspase activation in esophageal squamous cell carcinoma [68]. However, SFN can induce autophagy, probably through activation of the NRF2 pathway. Chloroquine added to SFN neutralizes NRF2 activation and increases caspase activation and apoptosis. In two previous studies [67,68], the onset of autophagy prevented apoptotic cell death, and the inhibition of autophagy enabled its onset.

In addition to the well-known resistance through the efflux activity of the ABCB1 transporter, its expression may also induce an additional reduction in cell sensitivity to substances that are not its substrate (reviewed in Reference [6]). After the expression of a mutated variant of ABCB1 incapable of vincristine transport, the cells showed resistance to this vinca alkaloid [69]. These effects may be related to the co-expression of ABCB1 with Bcl-2 family antiapoptotic proteins, as discussed above. However, this additional resistance associated with ABCB1 expression appears to be ruled out by multilayer mechanisms, which include changes in the regulation of LC3B-dependent autophagy described in the present study, upregulation of GRP78/BiP and its downstream processes [35,70], and changes in protein N-glycosylation and others [25,71] or in cytochrome P450 (2J6) expression [70].

## 4. Materials and Methods

### 4.1. Chemicals

SFN (1-Isothiocyanato-4-(methylsulfinyl)-butane) and glucoraphanin potassium salt (4-methylsulfinylbutyl glucosinolate potassium salt) were obtained from Sigma-Aldrich (MERCK spol. s r.o., Bratislava, Slovakia), unless otherwise stated in the text. All chemicals were from MERCK and were of analytical grade.

### 4.2. Cell Culture and Culture Conditions

The murine leukemic cancer cell line L1210 was obtained from Leibniz-Institut DSMZ-Deutsche Sammlung von Mikroorganismen und Zellkulturen GmbH (Braunschweig, Germany). We used three variants of the mouse lymphocytic leukemia cell line L1210: (i) ABCB1-negative drug-sensitive parental L1210 cells (ACC-123, S); (ii) ABCB1-positive, drug-resistant cells (R) overexpressing ABCB1 due to selection with vincristine [24]; and (iii) ABCB1-positive, drug-resistant cells (T) overexpressing ABCB1 due to stable transfection with Addgene plasmid 10957 (pHaMDRwt), and a retrovirus encoding full-length ABCB1 cDNA [72]. Transfection and cell characterization were completed as described elsewhere [25]. Cells were incubated in RPMI 1640 medium with L-glutamine (1 mg/mL) supplemented with 4% fetal bovine serum and 1 μg/mL gentamycin (all purchased from Gibco, Langley, OK, USA), at 37 °C, in a humidified atmosphere with 5% CO_2_. Cell viability was monitored by using a CASY Model TT-Cell Counter (Roche Applied Sciences, Madison, WI, USA). All cell variants (S, R and T) were cultivated in the absence or presence of SFN at an appropriate concentration and were used for further examination.

### 4.3. Cell Passage in the Presence or Absence of SFN

S, R and T cells (5 × 10^5^ cells/mL) were cultured in SFN-containing medium (concentration range 0–15 μM) in 6-well culture plates (5 mL per well) for 48 h. We considered this procedure to be a separate passage. Cell viability was monitored at various time intervals (4, 8, 12, 24 and 48 h) by measuring the plasma membrane integrity of individual cells through changes in electrical resistance induced by the cells that passed through the detector in the CASY Model TT-Cell Counter according to the manufacturer’s protocol. The time course of cell growth over 48 h under these conditions gives a straight line in the semilogarithmic plot that we used to evaluate changes in proliferation rate [70]. After the first passage, the cells were (i) used for analyses, (ii) seeded (5 × 10^5^ cells/mL) for another 48 h of culturing in fresh SFN-free medium and used for analyses and (iii) subjected to either 3 additional passages in 0–15 μM SFN medium or 5 additional passages in 0–10 μM SFN medium and used for analyses.

### 4.4. Determination of IC_50_ Values for S, R and T Cells after Each Passage in the Presence of SFN

S, R and T cells (5 × 10^5^ cells/mL) were repeatedly passaged up to 6 times in medium containing SFN (at concentrations 0.0, 2.5, 5.0, 7.5, 10.0 and 15.0 µM) in 6-well culture plates (5 mL per well). After 48 h, the number of viable cells was determined by using a CASY Model TT-Cell Counter according to the manufacturer’s protocol. Then 5 × 10^5^ cells/mL were subjected to further passage in fresh medium containing the appropriate concentration of SFN. The number of cells after passage in the absence of SFN was arbitrarily chosen to be 100%, and the other values were expressed relative to this value. The dependence of the number of viable cells as a function of ITC concentration in individual passages was fitted according to Equation (1), by nonlinear regression, using SigmaPlot 8.0 software (Systat Software. Inc., San Jose, CA, USA):(1)$$N\left(\%\right)=100\left(\%\right)\times \exp[\ln\left(0.5\right)\times {\left(\frac{c}{IC_{50}}\right)}^{n}]$$
where *N* in % is the number of viable cells after passaging in medium containing SFN at concentration, *c*; 100(%) is the number of viable cells after passaging in the absence of SFN; IC_50_ is the median inhibitory concentration of the respective ITC; and n is a coefficient that, in these experiments, reached a value of 0.89–1.11, i.e., close to one, and was therefore not taken into account. The validity of this equation was proven previously [23].

### 4.5. Cell Metabolic Activity Estimation Using the MTT Assay

Cell metabolic activity was assessed in terms of the NADH- and NADPH-dependent reduction of MTT (3-(4,5-dimethylthiazol-2-yl)-2,5-diphenyl tetrazolium bromide) to insoluble formazan. S cells (5 × 10^4^/well) were seeded in 0.2 mL medium with different concentrations of GL, SFN or a mixture of GL and recombinant enzyme (the preparation of which is described in References [27,47]) MYR (1 mU, 16.7 pkat of MYR per 1 nmol of GL). The noncompetitive activator AA was added to the medium at a concentration of 10 μM. After 48 h, the medium was replaced with FBS-free medium; 0.01 mL of MTT solution (4 mg/mL) was then added; and the cells were incubated for 3 h, at 37 °C, in the dark. After this time, the medium was removed, and the formazan crystals were dissolved in DMSO. The absorbance was measured at 540 nm, using a Universal Microplate Spectrophotometer µQuant (BioTek Instruments Inc., Winooski, VT, USA). Experiments were performed in octuplicates.

### 4.6. Estimation of Gene Expression by RT-PCR

S, R and T cells were differently passaged in SFN (see Section 4.3) and were used to determine the expression of the *Abcb1*, *Bax*, *Bcl2*, *Ccnb1*, *Ccnd1*, *Ccne1*, *Cdk2* and *Cdk6* genes. The *Gapdh* gene was used as an internal standard. After the appropriate passage, cells were harvested by centrifugation (664× *g* at 20 °C) and washed twice in PBS. Total RNA was isolated from cells by using TRI Reagent^®^ (MRC, Cincinnati, OH, USA) according to the manufacturer’s instructions. RNA samples were then quantified by using a NanoDrop instrument (λ_A260/280_). Reverse transcription (RT) was performed with 1 μg of total RNA, using Fermentas RevertAid Reverse Transcriptase for cDNA synthesis (Thermo Fisher Scientific, Waltham, MA, USA).

The polymerase chain reaction (PCR) was performed in a total volume of 25 μL, using a PCR kit according to the manufacturer’s protocol (Thermo Fisher Scientific). The PCR mixture contained 1 μL of cDNA obtained from the RT of total isolated RNA. The structures of the PCR primers for the respective genes are given in Table 3. PCR was performed by using the following timing: 30 cycles of 30 s of denaturation at 95 °C, 40 s of annealing at 57–59 °C and 1 min final extension at 72 °C. PCR products were separated on a 1.5% agarose gel (Lonza, Rockland, ME, USA) and visualized by GelRed^®^ nucleic acid gel staining (Biotium, Fremont, CA, USA). The primer sequences are documented in Table 3. Negligible expression of the murine *Abcb1* gene in S and T cells, as well as either its extensive expression in R cells or extensive expression of the human *ABCB1* gene in T cells, was continuously controlled.

### 4.7. Western Blot Detection of Cyclin B1, E1 and LC3B Protein Fragments

Protein was extracted from cultured cells by using RIPA lysis buffer containing 50 mM Tris-Cl (pH 8.0), 1% Triton X-100, 0.5% sodium deoxycholate, 0.1% SDS, 150 mM NaCl and protease inhibitor cocktail (Sigma, Saint Louis, MO, USA). The proteins from the samples were separated by sodium dodecyl sulfate–polyacrylamide electrophoresis, using 12% polyacrylamide (SDS-PAGE) gels, according to the protocol published by Laemmli [73]. The proteins were transferred by electroblotting onto nitrocellulose membranes (GE Healthcare Europe GmbH, Vienna, Austria), using the Towbin protocol [74,75]. The membranes were blocked for 1 h in 5% defatted milk solution in PBS-Tween 20. The primary antibodies were against: LC3B (affinity isolated rabbit polyclonal antibody L7543, Sigma-Aldrich, Saint Louis, MO, USA, which was diluted 1:300); against cyclin B1 (mouse monoclonal IgG1 antibody GNS1, Santa Cruz Biotechnology, Dallas, TX, USA, diluted 1:100); cyclin E1 (mouse monoclonal IgG2b antibody E-4, Santa Cruz Biotechnology, Dallas, TX, USA, diluted 1:100); and GAPDH (clone 6C5, MAB374, EMD Millipore Chemicals, Billerica, USA, diluted 1:500). Primary antibodies were added to blocking buffer containing 2.5% nonfat milk and allowed to interact with proteins on the Western blot membrane overnight at 4 °C. Mouse IgG kappa binding protein (m-IgGκ BP-HRP) and mouse anti-rabbit IgG-HRP, both conjugated with horseradish peroxidase (HRP), were used at a dilution of 1:1000 as secondary antibodies and were purchased from Santa Cruz Biotechnology. HRP signals were visualized by using an ECL detection system (GE Healthcare Europe GmbH, Vienna, Austria) on an Amersham Imager 600 (GE Healthcare Europe GmbH, Pittsburgh, PA, USA).

### 4.8. Visualization of Autophagic Vacuoles by MDC

The autofluorescent lysosomotropic compound MDC was recently introduced as a specific autophagolysosome marker to analyze the autophagic process [41]. After 4 passages in the absence or presence of SFN (5, 10 and 15 μM), S, R and T cells (5 × 10^5^) were resuspended in 200 µL serum-free medium containing 500 nM TQR (SelleckChem, Houston, TX, USA), and the cells were incubated at 37 °C for 45 min in the dark. After incubation, 50 µM MDC was added to cells with subsequent incubation for 1 h, at 37 °C, in the dark. The cells were washed with PBS, and the pellet was resuspended in 100 µL of PBS containing WGA conjugated with Texas Red (Invitrogen, Thermo Fisher Scientific, Eugene, OR, USA) at a final concentration of 1 µM for cell surface labeling. MDC was visualized by excitation with a 5% UV diode at 405 nm, and emitted fluorescence in the range of 515–574 nm was registered. WGA conjugated with Texas Red was visualized by excitation with a 7% white laser at 589 nm, and emitted fluorescence in the range 599–661 nm was registered. Imaging was performed on a Leica TCS SP8 AOBS confocal microscope (Leica Microsystems, Wetzlar, Germany) with an objective HC PL APO CS2 63 x/1.40 OIL.

### 4.9. Statistical Analysis and Data Processing

Numerical data are expressed as the mean ± SD. Statistical significance was assessed by unpaired Student’s *t*-test, using SigmaPlot 8.0 software (Systat Software, Inc., San Jose, CA, USA).

## 5. Conclusions

Here, we investigated the effect of SFN during repeated cultivations of leukemia cells L1210 in relation to the expression of the ABCB1 transporter. First, we observed that, even though SFN suppresses cell growth in a concentration-dependent manner, cells after a single culture in the presence of SFN are able to intensively proliferate in a medium without SFN in the subsequent culturing. Repeated cell cultures, depending on the number of repetitions, accentuated cell damage and caused a decrease in the number of viable cells. This is more pronounced in ABCB1-negative (S) cells than in ABCB1-positive R and T cells. Sulforaphane caused the onset of autophagy in L1210 cells, accompanied by an increase in the content of the 16 and 18 kDa fragments of LC3B protein and the enhancement of the specific staining of autophagic vesicles by monodansylcadaverine. Autophagy was less developed in ABCB1-positive R and T cells than in their ABCB1-negative counterparts. We conclude that ABCB1-positive cells have a slightly reduced sensitivity to SFN when it is repeatedly used on them; however, we do not assume that SFN is removed from the cells by this transporter. This may also be due to changes in the formation and processing of the LC3B protein and the progression of autophagy. Other changes in the expression of regulatory proteins such as cyclins and CDKs and Bcl-2 may also influence the response of cells to repeated cultures in the presence of SFN.

## Figures and Tables

**Figure 1 ijms-23-10818-f001:**
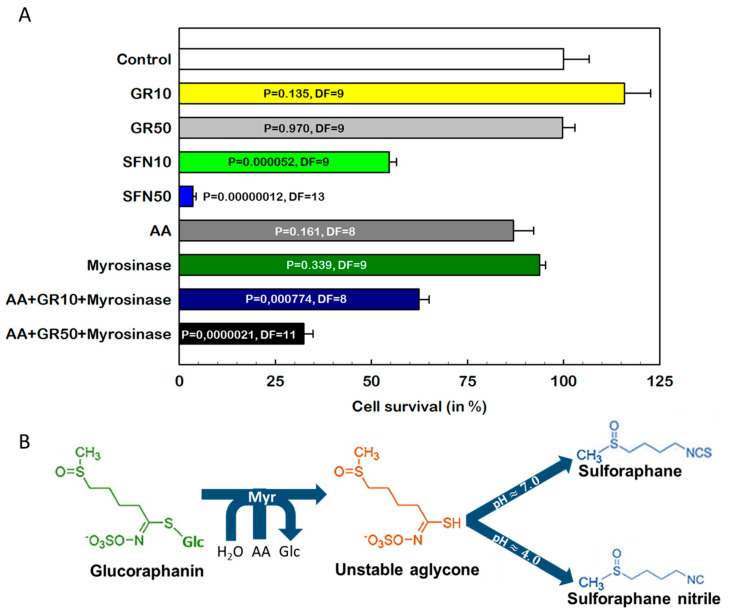
Comparison of sulforaphane (SFN) and glucoraphanin (GR) in the induction of S-cell death. Glc, glucose; Myr, myrosinase; AA, ascorbic acid. (**Panel A**) Cell death induced by SFN and GR alone or in combination with Myr and AA in S cells. S cells were cultured under standard conditions with either GR (10 μM and 50 μM), SFN (10 and 50 μM), AA (10 μM) and recombinant Myr, each alone, or with GR in combination with Myr and AA at the same concentration. After incubation of cells, the MTT assay was used to estimate cell viability. The results represent the mean values ± SD. Changes were considered significant when *p* < 0.05. (**Panel B**) Scheme of GR activation by the myrosinase reaction.

**Figure 2 ijms-23-10818-f002:**
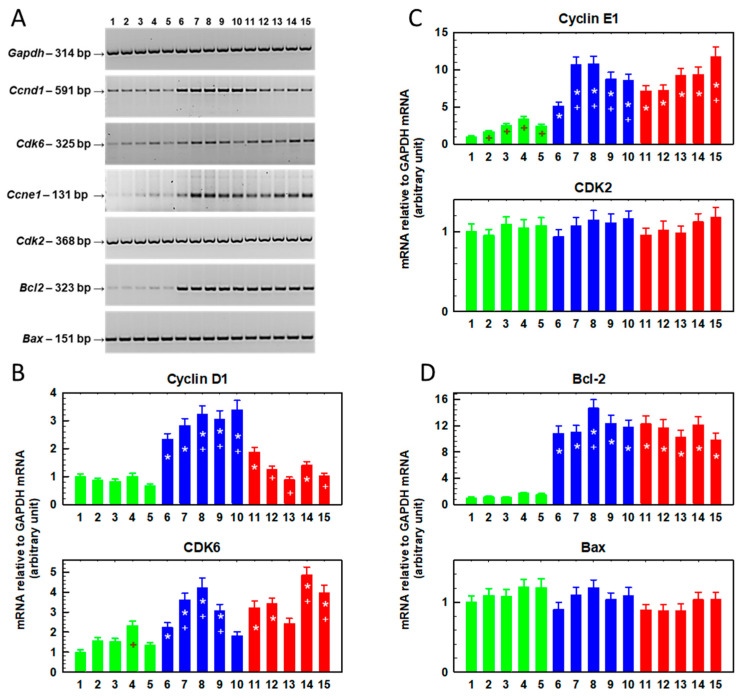
Changes in the relative gene expression of *Ccnd1* (for cyclin D1), *Cdk6* (for cyclin-dependent kinase 6), *Ccne1* (for cyclin E1), *Cdk2* (for cyclin-dependent kinase 2), *Bcl2* and *Bax* in S, R and T cells passaged once in SFN-containing medium. The cells were cultured for 48 h in the presence of SFN at the following concentrations: (1) 0.0 μM S cells, (2) 2.5 μM S cells, (3) 5.0 μM S cells, (4) 7.5 μM S cells, (5) 10.0 μM S cells, (6) 0.0 μM R cells, (7) 2.5 μM R cells, (8) 5.0 μM R cells, (9) 7.5 μM R cells, (10) 10.0 μM R cells, (11) 0.0 μM T cells, (12) 2.5 μM T cells, (13) 5.0 μM T cells, (14) 7.5 μM T cells and (15) 10.0 μM T cells. (**Panel A**) Electrophoretic analysis of the respective PCR products. The *GAPDH* gene was used as an internal control. Data are representative of three independent measurements. (**Panels B**–**D**) The optical densities of the PCR product bands were quantified and are summarized in bar plots. The data are expressed as the mean ± SD of three independent measurements. S cells (green), R cells (blue) and T cells (red). Significance * means that data differ from corresponding conditions observed in S cells at the level *p* < 0.02; + means that data differ from corresponding conditions observed for cells in the absence of SFN *p* < 0.02.

**Figure 3 ijms-23-10818-f003:**
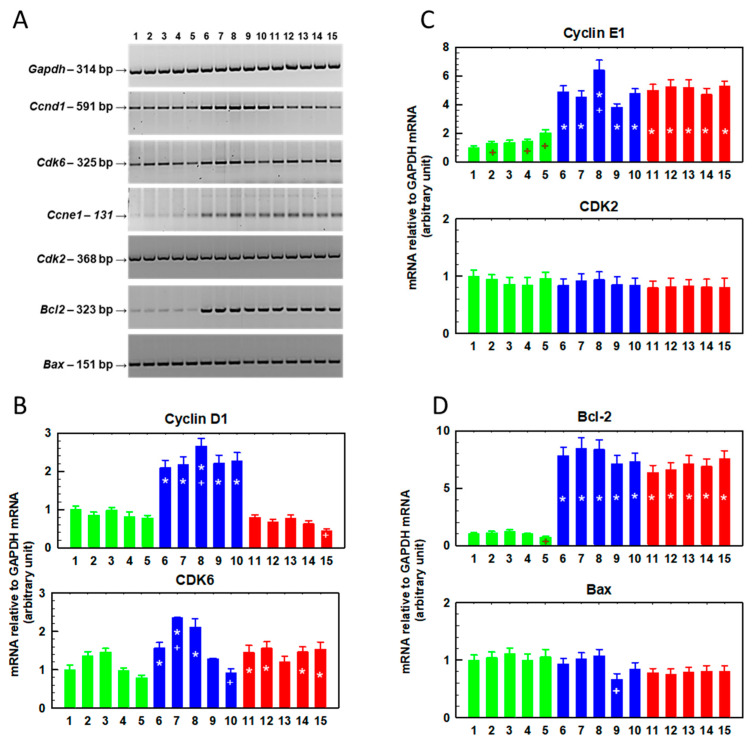
Changes in the relative gene expression of *Ccnd1* (for cyclin D1), *Cdk6* (for cyclin-dependent kinase 6), *Ccne1* (for cyclin E1), *Cdk2* (for cyclin-dependent kinase 2), *Bcl2* and *Bax* in S, R and T cells cultured once in SFN-containing medium followed by further cultivation in SFN-free medium. The cells were cultured for 48 h in the presence of SFN at the following concentrations: (1) 0.0 μM S cells, (2) 2.5 μM S cells, (3) 5.0 μM S cells, (4) 7.5 μM S cells, (5) 10.0 μM S cells, (6) 0.0 μM R cells, (7) 2.5 μM R cells, (8) 5.0 μM R cells, (9) 7.5 μM R cells, (10) 10.0 μM R cells, (11) 0.0 μM T cells, (12) 2.5 μM T cells, (13) 5.0 μM T cells, (14) 7.5 μM T cells and (15) 10.0 μM T cells. After this time period, the cells were cultured for another 48 h in the absence of SFN. (**Panel A**) Electrophoretic analysis of the respective PCR products. The *GAPDH* gene was used as an internal control. Data are representative of three independent measurements. (**Panels B**–**D**) The optical densities of the PCR product bands were quantified and are summarized in the bar plots. The data are expressed as the mean ± SD of three independent measurements. S cells (green), R cells (blue) and T cells (red). Significance * means that the data differ from corresponding conditions observed for S cells at the level *p* < 0.02; + means that the data differ from corresponding conditions observed for cells in the absence of SFN *p* < 0.02.

**Figure 4 ijms-23-10818-f004:**
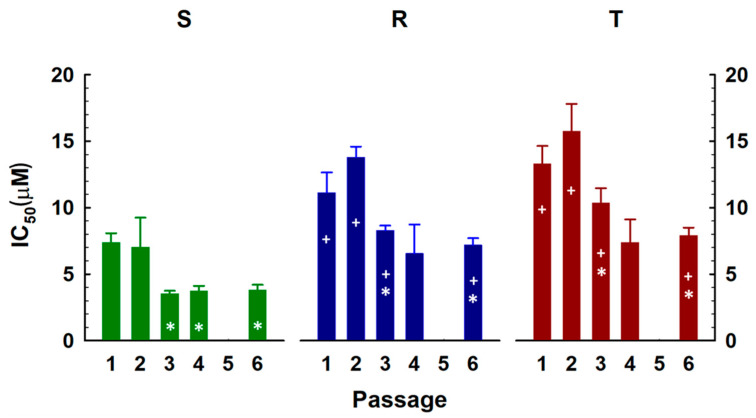
IC_50_ changes induced by repeated cultivations in SFN-containing medium. Data were calculated by nonlinear regression of the dose-response curves according to Equation (1). Data represent calculated SD values for 44 degrees of freedom. Significance: + data differ from S cells at the levels *p* < 0.02; * data differ from control at the absence of SFN at the level *p* < 0.02.

**Figure 5 ijms-23-10818-f005:**
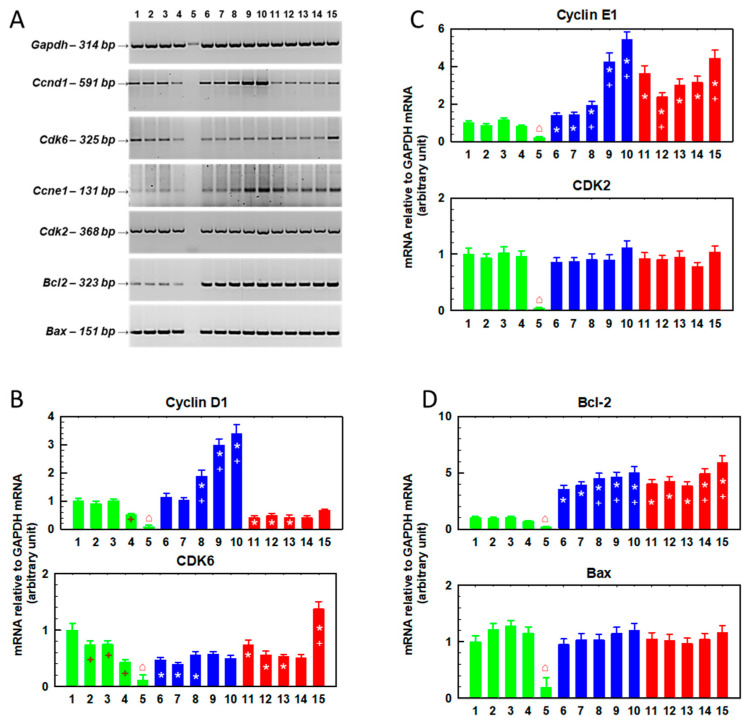
Changes in the relative gene expression of *Ccnd1* (for cyclin D1), *Cdk6* (for cyclin-dependent kinase 6), *Ccne1* (for cyclin E1), *Cdk2* (for cyclin-dependent kinase 2), *Bcl2* and *Bax* in S, R and T cells passaged six times in SFN-containing medium. At each passage, cells were cultured for 48 h in the presence of SFN at the following concentrations: (1) 0.0 μM S cells, (2) 2.5 μM S cells, (3) 5.0 μM S cells, (4) 7.5 μM S cells, (5) 10.0 μM S cells, (6) 0.0 μM R cells, (7) 2.5 μM R cells, (8) 5.0 μM R cells, (9) 7.5 μM R cells, (10) 10.0 μM R cells, (11) 0.0 μM T cells, (12) 2.5 μM T cells, (13) 5.0 μM T cells, (14) 7.5 μM T cells and (15) 10.0 μM T cells. (**Panel A**) Electrophoretic analysis of the respective PCR products. The *GAPDH* gene was used as an internal control. Data are representative of three independent measurements. The number of S cells after six passages in the presence of SFN at a concentration of 10 μM was significantly reduced, and after isolation of total RNA, we did not obtain enough quality polyribonucleic acid for GAPDH and other genes detection (⌂). (**Panels B**–**D**) The optical densities of the PCR product bands were quantified and are summarized in bar plots. The data are expressed as the mean ± SD of three independent measurements. S cells (green), R cells (blue) and T cells (red). Significance * means that the data differ from corresponding obtained for S cells at the level *p* < 0.02; + means that the data differ from corresponding conditions observed for cells in the absence of SFN *p* < 0.02.

**Figure 6 ijms-23-10818-f006:**
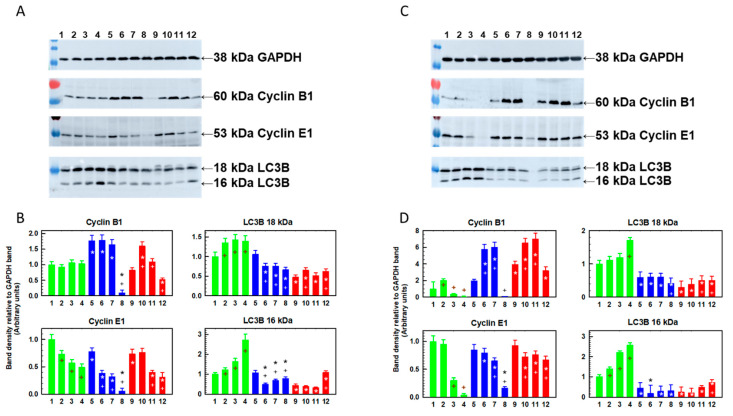
Changes in the relative content of cyclin B, cyclin E and LC3B protein fragments (16 and 18 kDa) in S, R and T cells passaged either one or four times in SFN-containing medium. Panels A and B: The cells were cultured for 48 h in the presence or absence of SFN. Panels C and D: Cells were passaged four times for 48 h in the presence or absence of SFN. SFN concentrations: (1) 0 μM S cells, (2) 5 μM S cells, (3) 10 μM S cells, (4) 15 μM S cells, (5) 0 μM R cells; (6) 5 μM R cells, (7) 10 μM R cells, (8) 15 μM R cells, (9) 0 μM T cells; (10) 5 μM T cells, (11) 10 μM T cells and (12) 15 μM T cells. (**Panels A,C**) Western blot determination of the respective proteins. Signals for GAPDH were used as internal controls. Data are representative of three independent measurements. (**Panels B,D**) The optical densities of the protein bands were quantified and are summarized in bar plots. The data are expressed as the mean ± SD of three independent measurements. S cells (green), R cells (blue) and T cells (red). Significance * means that the data differ from corresponding conditions observed for S cells at the level *p* < 0.02; + means that data differ from corresponding obtained for cells in the absence of SFN *p* < 0.02.

**Figure 7 ijms-23-10818-f007:**
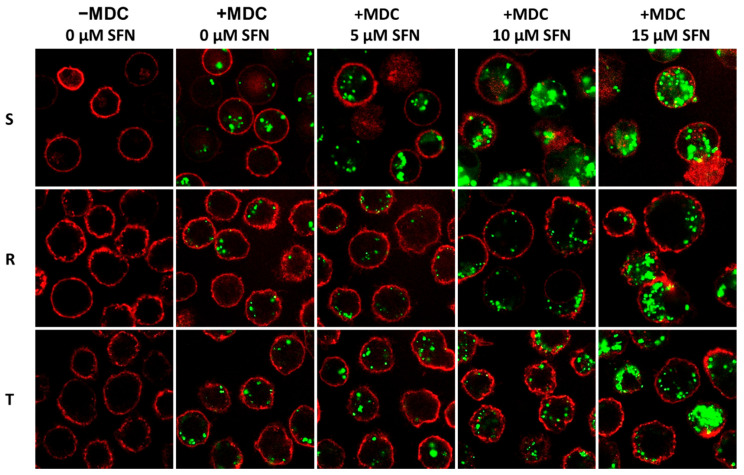
Autophagic vesicles after four passages of S, R and T cells in medium containing SFN at the indicated concentration were labeled MDC (green fluorescence). MDC was added in the presence of 500 nM TQR. The data are representative of three independent experiments. Wheat-germ agglutinin (WGA) conjugated with Alexa Fluor™ 647 was used to label the cell surface. This lectin marks the surface of S, R and T cells [43,44].

**Table 1 ijms-23-10818-t001:** Summary of changes in the expression of the indicated genes after different passage procedures of S, R and T cells in the presence of SFN.

One passage in the presence of SFN																		
Gene	*Ccnd1*			*Cdk6*			*Ccne1*			*Cdk2*			*Bcl2*			*Bax*		
Cell variants	S	R	T	S	R	T	S	R	T	S	R	T	S	R	T	S	R	T
Compared to S		↑	0		↑	↑		↑	↑		0	0		↑	↑		0	0
SFN (μM)	Compared to cells passaged in the absence of SFN																	
2.5																		
5.0																		
7.5																		
10.0																		
One passage in the presence of SFN followed by passage in the absence of SFN																		
Gene	*Ccnd1*			*Cdk6*			*Ccne1*			*Cdk2*			*Bcl2*			*Bax*		
Cell variants	S	R	T	S	R	T	S	R	T	S	R	T	S	R	T	S	R	T
Compared to S		↑	0		↑	↑		↑	↑		0	0		↑	↑		0	0
SFN (μM)	Compared to cells passaged in the absence of SFN																	
2.5																		
5.0																		
7.5																		
10.0																		
Six consecutive passages in the presence of SFN																		
Gene	*Ccnd1*			*Cdk6*			*Ccne1*			*Cdk2*			*Bcl2*			*Bax*		
Cell variants	S	R	T	S	R	T	S	R	T	S	R	T	S	R	T	S	R	T
Compared to S		0	** ↓ **		** ↓ **	** ↓ **		↑	↑		0	0		↑	↑		0	0
SFN (μM)	Compared to cells passaged in the absence of SFN																	
2.5																		
5.0																		
7.5																		
10.0																		

The 0 means that the corresponding value is not different from the value obtained for S cells; ↑ means that the corresponding value is significantly higher (*p* < 0.02) than the value obtained for S cells; ↓ means that the corresponding value is significantly lower (*p* < 0.02) than the value obtained for S cells;        means that the passage in the presence of SFN at a given concentration does not induce changes in gene expression;         means that the passage in the presence of SFN at a given concentration induces a significant (*p* < 0.02) increase in gene expression;       means that the passage in the presence of SFN at a given concentration induces a significant (*p* < 0.02) reduction in gene expression.

**Table 2 ijms-23-10818-t002:** Summary of changes in the cell content of the indicated protein after different passage procedures of S, R and T cells in the presence of SFN.

One passage in the presence of SFN												
Proteins	Cyclin B1			Cyclin E1			LC3B					
							16 kDa			18 kDa		
Cell variants	S	R	T	S	R	T	S	R	T	S	R	T
Compared to S		↑	0		** ↓ **	** ↓ **		0	** ↓ **		0	** ↓ **
SFN (μM)	Compared to cells passaged in the absence of SFN											
5												
10												
15												
Four passages in the presence of SFN												
Proteins	Cyclin B1			Cyclin E1			LC3B					
							16 kDa			18 kDa		
Cell variants	S	R	T	S	R	T	S	R	T	S	R	T
Compared to S		0	↑		0	0		** ↓ **	** ↓ **		** ↓ **	** ↓ **
SFN (μM)	Compared to cells passaged in the absence of SFN											
5												
10												
15												

The 0 means that the corresponding value is not different from the value obtained for S cells; ↑ means that the corresponding value is significantly higher (*p* < 0.02) than the value obtained for S cells; ↓ means that the corresponding value is significantly lower (*p* < 0.02) than the value obtained for S cells;        means that the passage in the presence of SFN at a given concentration does not induce changes in protein content;         means that the passage in the presence of SFN at a given concentration induces a significant (*p* < 0.02) increase in protein content;       means that the passage in the presence of SFN at a given concentration induces a significant (*p* < 0.02) reduction in protein content.

**Table 3 ijms-23-10818-t003:** PCR primers for estimation of respective genes.

Gene	Primer Sequences	T_A_ (°C)	PCR Product (bp)
*ABCB1* *	F: 5′-GCAATGGAGGAGCAAAGAAG-3′	59	150
	R: 5′-CCAAAGTTCCCACCACCATA-3′		
*Abcb1*	F: 5′-AGGTAGAGACACGTGAGGTCG-3′	57	158
	R: 5′-CAGCCAACCTGCATAACG-3′		
*Bax*	F: 5′-CTAGCAAAGTAGAAGAGGGCAACC-3′	58	151
	R: 5′-ATGAACTGGACAGCAATATGGAG-3′		
*Bcl2*	F: 5′-GCATGCTGGGGCCATATAGTT-3′	58	323
	R: 5′-GGCTGGGGATGACTTCTCTC-3′		
*Ccnb1*	F: 5′-GGTGACTTCGCCTTTGTGAC-3′	58	125
	R: 5′-CTACGGAGGAAGTGCAGAGG-3′		
*Ccnd1*	F: 5′-GGCCTTCAGGCAAAAACCAG-3′	58	591
	R: 5′-TCACCCTGAGAGTAGGGAGC-3′		
*Ccne1*	F: 5′-AGGATGACGCTGCAGAAAGT-3′	58	131
	R: 5′-GGAAAATCAGACCACCCAGA-3′		
*Cdk2*	F: 5′-CTTTGCTGAAATGGTGACCCG-3′	57	368
	R: 5′-CCAGGGCCAAGTCAGACCAC-3′		
*Cdk6*	F: 5′-GCCGGGCTCTGGAACTTTAT-3′	57	325
	R: 5′-CGCCGATCAGCAGTATGAGT-3′		
*Gapdh*	F: 5′-CAATGTGTCCGTCGTGGAT-3′	57	314
	R: 5′-GTGGGTGGTCCAGGGTTT-3′		

* Primers for human gene, all other primers are for mouse genes.

## Data Availability

Additional data, as well as resistant variants of L1210 cells, are available from the authors.

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
