# Peer review of "Effects of Sulforaphane-Induced Cell Death upon Repeated Passage of Either P-Glycoprotein-Negative or P-Glycoprotein-Positive L1210 Cell Variants"

_ijms, 2022, doi:10.3390/ijms231810818_

Round 1

Reviewer 1 Report

In this manuscript, Bertova et al., studied the “Effects of sulforaphane-induced cell death upon repeated passage of three variants of murine leukemic cancer cell line L1210 with different level expression of ABCB1. The work is interesting, and the manuscript is well written.

11.  For cell gene expression experiments, it would be an interesting to see the activation of Nrf2 is different in the cell types (S, R, T) under repeated SFN treatments.

2 2.  The most significant differences between ABCB1-negative S cells and ABCB1-positive R cells were observed for the cell levels of LC3B protein fragments. This is interesting. What about P62 levels (changes) in these two cell types?

Author Response

Thank you for your comments which improve the quality of our article

Albert Breier

on behalf of all authors

Reviewer 2 Report

In the manuscript ijms-1908845 titled “Effects of sulforaphane-induced cell death upon repeated passage of either P-glycoprotein-negative or P-glycoprotein-positive L1210 cell variants”, authors have developed an experimental approach to test the effect of SFN repeated passage of in ABCB1-negative (S) and ABCB1-positive variants of L1210 cells (R and T).

While the developed cell-based assays and the workflow is relevant for understanding and addressing the gap in knowledge in the field, and the authors’ conclusions are well supported with the results, I believe authors should address the minor comments below before publishing the manuscript.

Minor comments

1)     It is not clear why authors have randomly selected the concentrations of 10 µM and 50 µM SFN to test its effect on cell death. Even though there is evidence for the concentration dependence of Ascorbic acid, the correlation between SFN concentration has not been investigated. Please add a justification for using these concentrations (Is this concentration of SFN of biological relevance?)

2)     In Figure 1 the conditions for the control experiment are not very clear. Since it seems the cell growth is higher in the presence of   10 µM GR in comparison to the controls.

3)     How is the pH of the medium maintained at pH=7 to obtain the major product of SFN by the reaction of Glucoraphanin with myrosinase and ascorbic acid, since Ascorbic acid provides an acidic nature to the medium? Will this have an affect on the observed data?

Considering the soundness of the scientific approach and findings, I would enthusiastically recommend the manuscript for publication after authors addressing the suggested minor revisions. I believe addressing these comments will improve the manuscript and make the readers understand better.

Author Response

Thank you for your comments, which improve the quality of our article

Albert Breier

on behalf of all authors
